# Perceptions of COVID-19-related nudges in the Arab world: A cross-country analysis of approval rates and associated factors

Fadi Makki[1,2©], Belal Nedal Sabbah[3©], Hani Tamim[3,4], Mariam Abdelnabi[1,5], Paola Schietekat[6], Nabil Saleh[7], Ali Osseiran[1], Abdulkarim Almakadma[3,8], Mohamed Al-Komi[9], Ebaa Alsayed[9], Rajaa Fakhoury[3,10], Fatima Saleh[10], Basema Saddik[11,12,13], Hana M. A. Fakhoury[3*], Sarah Daher[3], Cass R. Sunstein[14]

**1** Nudge Lebanon, Beirut, Lebanon, **2** The Behavioral Science Lab, Boston Consulting Group, Washington DC, United States of America, **3** College of Medicine, Alfaisal University, Riyadh, Saudi Arabia, **4** Clinical Research Institute, Department of Internal Medicine, American University of Beirut Medical Center, Beirut, Lebanon, **5** Warwick Business School, University of Warwick, Coventry, United Kingdom, **6** Neovantas Consulting, Madrid, Spain, **7** NL Partners, Toronto, Canada, **8** College of Medicine, Qatar University, Doha, Qatar, **9** Behavioral and Economic Decision-Making Lab, School of Business, American University in Cairo, Cairo, Egypt, **10** Faculty of Health Sciences, Beirut Arab University, Beirut, Lebanon, **11** Department of Public Health and Epidemiology, College of Medicine and Health Sciences, Khalifa University, Abu Dhabi, United Arab Emirates, **12** College of Medicine, University of Sharjah, Sharjah, United Arab Emirates, **13** School of Population Health, Faculty of Medicine and Health, University of New South Wales, Sydney Australia, **14** Harvard University, Cambridge, Massachusetts, United States of America

© The authors contributed equally to this work and share first authorship
* hana.fakhoury@gmail.com

## Abstract

The COVID-19 pandemic has necessitated novel approaches to influence public behavior. While "nudging" has gained prominence in Western contexts, its perception and effectiveness in the Arab world remain understudied. This study aimed to investigate the approval of COVID-19-related nudges across four Arab countries and explore associated sociodemographic factors. A cross-sectional study was conducted from November 2020 to January 2022, involving 698 participants from Egypt, Lebanon, Saudi Arabia, and the United Arab Emirates. Participants were presented with eight hypothetical COVID-19-related nudges categorized according to distinct behavioral mechanisms: choice architecture (e.g., floor markers, prominent placement of fruits and vegetables), information disclosure (publicly sharing infection causes), moral appeals (letters from elderly urging compliance), social norm enforcement (public shaming of violators and use of spoilers on billboards), and surveillance-based interventions (GPS tracking of quarantined individuals). Approval rates varied widely (50%–95%). Less intrusive nudges received the highest support: supermarket floor markers (95.4%), prominent display of fruits and vegetables (88.8%), park area divisions (82.0%), infection cause disclosure (86.5%), and elderly letters urging compliance (84.1%). Approval was lower for more intrusive measures,

**Data availability statement:** The data sets used and/or analyzed during the current study are available on an online repository through this link: https://osf.io/eav4j/.

**Funding:** The authors received no specific funding for this work.

**Competing interests:** The authors have declared that no competing interests exist.

including billboard spoilers (52.0%) and public shaming of curfew violators (49.9%). GPS tracking, the most intrusive intervention, received intermediate approval (72.8%). Higher COVID-19 concern was significantly associated with greater approval of nudges (p < 0.001), with age, gender, and family COVID-19 status also influencing approval rates. These findings demonstrate generally positive attitudes towards COVID-19-related nudges among university affiliates in four Arab countries, with clear variations according to nudge type, intrusiveness, and sociodemographic characteristics. While the results offer valuable insights for culturally tailored behavioral interventions in the Arab world, they reflect a university setting and may not be generalizable to the broader public.

## Introduction

The COVID-19 pandemic has presented unprecedented challenges to public health systems worldwide, necessitating rapid and effective behavioral interventions to curb the spread of the virus [1]. In this context, the concept of "nudging," a behavioral science approach that subtly guides decision-making without restricting choice, has gained prominence as a potential tool for policymakers [2]. Nudges have been successfully employed in various domains, including health, finance, and environmental conservation [3–5].

The effects of nudging have been well-documented in various Western contexts; however, empirical evidence indicates that its impact varies across different behavioral domains and settings. A comprehensive meta-analysis by Mertens et al. (2022) found that while nudging interventions generally produced positive effects, the magnitude of these effects varied significantly across different behavioral domains, ranging from small to moderate effect sizes (Cohen's d = 0.24 to 0.65) [6]. Notably, the financial domain exhibited the smallest effects, suggesting that nudges may be less effective in influencing financial behaviors.

Despite the rich literature on nudging in Western contexts, there is a notable gap in our understanding of its perception and efficacy in non-Western cultures, particularly in the Arab world. Cultural differences can significantly influence the acceptability and effectiveness of behavioral interventions [7]. Hofstede's cultural dimensions theory presents a valuable framework for this study [8]. It provides a structured lens to examine how cultural dimensions such as power distance, individualism, uncertainty avoidance, and indulgence may shape nudge acceptability in the Arab world, where collectivism and hierarchical social structures are prominent [9,10]. Although criticized [11], Hofstede's framework suits our aim of this study, which is to demonstrate how similar nudges can have different reactions depending on the differences in the cultural dimensions of societies [8].

Previous research has explored public attitudes towards nudges in various countries. Sunstein et al. found general approval for nudges across several European countries and the United States, with variations based on the type of nudge and cultural context [12]. Similarly, Jung and Mellers observed that Americans generally

approved of nudges, but were less supportive of those perceived as manipulative or coercive [13]. However, these studies primarily focused on Western populations, leaving a significant gap in our understanding of nudge perceptions in other cultural contexts.

The Arab world, with its distinct cultural, social, and political landscape, presents a unique setting for examining the acceptability of nudges. The region's collectivist culture, strong family ties, and varying degrees of government trust may influence how nudges are perceived and accepted [9,10]. Moreover, the COVID-19 pandemic has heightened the urgency of understanding effective behavioral interventions in this region, which has faced significant challenges in managing the outbreak [14].

This study aimed to address this knowledge gap by investigating the approval of COVID-19-related nudges across four Arab countries in the university setting. The target population comprised university affiliates, including students, faculty, and administrative staff, from the four participating institutions. We explored public sentiment towards eight distinct nudges, categorized into five types: choice architecture, information disclosure, moral appeals, social norm enforcement, and surveillance-based interventions. By examining the approval rates of these varied nudges, we sought to understand which types of interventions are most acceptable in Arab societies and how this acceptance varies across different demographic groups.

## Materials and methods

### Study design and participants

This cross-sectional study was conducted between 24 November 2020 and 5 January 2022 across four Arab countries: Egypt, Lebanon, Saudi Arabia (KSA), and the United Arab Emirates (UAE). The study population consisted of university affiliates (students, faculty, and administrative staff) from four institutions: the American University in Cairo (Egypt), Beirut Arab University (Lebanon), Alfaisal University (KSA), and the University of Sharjah (UAE). In total, 698 individuals participated by responding to an online survey distributed through institutional mailing lists.

### Data collection

Recruitment and data collection did not commence simultaneously across all study sites. Instead, each partner university launched its survey at different time points. In some institutions, the start of data collection was delayed to avoid conflicts with semester-end exams. On average, data collection in each country lasted approximately three months.

Alfaisal University was the first to initiate data collection, distributing the survey on November 24, 2020, and closing recruitment on February 5, 2021. The American University in Cairo followed, launching its survey on December 2, 2020, and concluding data collection on February 15, 2021. The University of Sharjah began data collection on February 4, 2021, and completed it on May 5, 2021. The final site, Beirut Arab University, launched its survey on September 27, 2021, and concluded recruitment on January 5, 2022.

Despite variations in recruitment timelines across sites, data analysis commenced only after all responses had been collected. Informed consent was obtained online through an electronic consent form presented at the beginning of the survey. Participants were informed that their participation was voluntary and that their responses would remain anonymous and confidential. Only those who actively indicated consent (by selecting a checkbox) were able to proceed with the survey. Anonymity was maintained by collecting only non-sensitive, non-identifiable demographic data, with no names, email addresses, or other personal identifiers recorded. To ensure confidentiality, access to the survey data was restricted to the principal investigators and the statistician involved in the analysis.

### Ethical considerations

The study received ethical approval from the Institutional Review Boards of all four participating universities: Alfaisal University Institutional Review Board, approval number IRB-20065; American University in Cairo (AUC) Research Ethics

Committee, approval number 2020-2021-005; Beirut Arab University (BAU) Institutional Review Board, approval number 2021-H-0122-HS-R-0442; and the University of Sharjah Research Ethics Committee, approval number REC-20-11-04-02.

### Inclusivity in global research

Additional information regarding the ethical, cultural, and scientific considerations specific to inclusivity in global research is included in the Supporting Information (S1 Checklist. Inclusivity in Global Research Checklist).

### Survey instrument

A standardized questionnaire was developed and administered via SurveyMonkey. Recruitment occurred via institutional mailing lists, with one initial invitation and two reminders during the recruitment period. The electronic survey was distributed via email to all university affiliates with participation being voluntary and no incentives provided. The survey was available in both Arabic and English and consisted of three main sections.

The first section gathered sociodemographic characteristics including age, gender, education level, occupation, income, marital status, and number of children.

The second section focused on health-related factors, including self-reported health status, measured by the question "How would you describe your current health?" on a 7-point Likert scale (1=very poor, 7=excellent), COVID-19 concern level, measured on a 7-point Likert scale (1=not at all concerned, 7=extremely concerned), and personal or family history of COVID-19 infection.

In the third section, participants were presented with eight hypothetical COVID-19-related nudges, each reflecting a distinct behavioral mechanism.

- Choice architecture nudges included: (1) social distancing floor markers in supermarkets; (2) park area divisions limiting the number of people per section (e.g., a maximum of 10 individuals per square), achieved by demarcating green spaces into marked areas; and (3) the prominent display of fruits and vegetables as the first items on grocery store mobile apps and websites.

- Information disclosure nudges included: (4) public disclosure of infection causes, referring to the reporting of aggregated, anonymized information (e.g., "because of the exchange of hugs and kisses at a family gathering, nine cases of COVID-19 have been detected, three of which required hospitalization"), as practiced by several Ministries of Health in the region, rather than the disclosure of identifiable individual data.

- Moral nudges included: (5) elderly people sending letters to family members urging them to comply with preventive rules.

- Social norm enforcement nudges included: (6) public shaming through publishing the nationalities of curfew violators and (7) the use of spoilers on billboards, involving the public display of plot details from popular television series during lockdowns to discourage non-essential travel.

- Surveillance-based interventions included: (8) GPS tracking of quarantined individuals via a mobile app.

For each scenario, participants indicated their approval or disapproval of the nudge using a binary (approve/disapprove) response format.

### Nudge selection

The nudges were selected to capture a diverse range of behavioral mechanisms relevant to the COVID-19 context and culturally salient in the Arab region. The set was curated from both implemented interventions and illustrative examples, including several directly inspired by measures adopted during the pandemic in Middle Eastern countries—such as GPS tracking, social distancing floor markers, and public disclosure of infection causes.

Selection was also intentional in covering a spectrum of intrusiveness, ranging from subtle strategies such as product placement in grocery apps (choice architecture) to more overt interventions like public shaming and billboard spoilers (social norm enforcement, with the latter relying on deterrence through aversive sanctioning). Although the prominent display of fruits and vegetables in grocery apps was not COVID-specific, it was included as a choice architecture nudge because several ministries of health in the region, along with the WHO Eastern Mediterranean Regional Office, promoted healthy eating during the pandemic to support immune function and reduce vulnerability to infection [15].

To ensure contextual appropriateness, we sought feedback from regional and behavioral science experts, who reviewed a preliminary list and provided informal input on perceived intrusiveness and pandemic relevance. Their feedback informed the refinement of the final set of interventions.

To systematically assess and classify the intrusiveness of the eight interventions, we applied the Nuffield Council on Bioethics Intervention Ladder [16], a widely used framework in public health ethics. This framework categorizes interventions according to the extent to which they limit individual autonomy, from least intrusive ("providing information") to most intrusive ("restricting choice"). Each intervention was mapped to a rung of the ladder to ensure transparent and consistent classification. For example, public disclosure of infection causes was classified as "providing information," supermarket floor markers and grocery app defaults as "changing the default," billboards spoiling TV shows as "utilizing disincentives," and GPS tracking as "restricting choice." A complete summary of how each intervention was mapped onto the Nuffield Intervention Ladder is provided in S1 Table.

## Statistical analysis

Descriptive statistics were used to summarize sociodemographic characteristics and nudge approval rates. Categorical variables were summarized by the number and percent, whereas the continuous ones were summarized by mean and standard deviation (SD) or median and interquartile range (IQR). Chi-square or Fisher's exact tests were used to assess the association between demographic and health related factors and nudge approval.

## Results

### Sociodemographic characteristics

Our study included 698 participants from four Arab countries. Responses by institution were: 106 from Alfaisal University (Saudi Arabia; 15.2%), 90 from the American University in Cairo (Egypt; 12.9%), 386 from Beirut Arab University (Lebanon; 55.3%), and 116 from the University of Sharjah (United Arab Emirates; 16.6%).Based on publicly available data regarding the size of each university community—including students, faculty, and staff—we estimate the response rates to be approximately: 3.9% for Alfaisal University (≈2,700 affiliates), 1.2% for AUC (≈7,600 affiliates), 2.2% for BAU (≈17,300 affiliates), and 0.5% for the University of Sharjah (≈23,100 affiliates). The sample was predominantly female (57.3%), and the median age was 22 years (IQR: 19–35). Due to skewness in the age distribution, the median with interquartile range (IQR) is reported instead of the mean and standard deviation. Students represented the majority of the participants (65.5%), while the rest consisted of academic/administrative staff (34.5%). Regarding education, 40.0% had secondary education or lower, 34.8% had an undergraduate degree, and 25.2% had a graduate degree. The majority of participants were single (73.9%) and from middle-income households (51.0%) (Table 1).

### Nudge approval rates

We observed a general trend of approval for the tested nudges, with approval ranging from 49.9% to 95.4%. However, support varied not only by nudge category but also by intrusiveness level (Table 2).

**Table 1. Sociodemographic and health-related characteristics of participants (N = 698).**

| Variables | | n (%) |
|---|---|---|
| University | Alfaisal | 106 (15.2%) |
| | AUC | 90 (12.9%) |
| | BAU | 386 (55.3%) |
| | Sharjah | 116 (16.6%) |
| Female | | 400 (57.3%) |
| Age | Median (IQR) | 22 (19 - 35) |
| Education | Secondary education or lower | 279 (40.0%) |
| | Undergraduate degree | 243 (34.8%) |
| | Graduate degree | 176 (25.2%) |
| Occupation | Student | 457 (65.5%) |
| | Academic Staff/Admin/Administrative Staff/Faculty | 241 (34.5%) |
| Major | Medicine | 197 (32.7%) |
| | Health | 55 (9.1%) |
| | Non-health | 351 (58.2%) |
| Income | Low | 117 (19.9%) |
| | Middle | 300 (51.0%) |
| | High | 171 (29.1%) |
| Marital status | Single | 516 (73.9%) |
| | Married | 182 (26.1%) |
| Number of children | Median (IQR) | 0 (0 - 1) |
| BMI categories (self-reported) | Underweight | 46 (7.6%) |
| | Healthy | 320 (53.0%) |
| | Overweight | 173 (28.6%) |
| | Obese | 65 (10.8%) |
| Self-reported health status | Mean ± SD | 5.5 ± 1.2 |
| | Median (IQR) | 6 (5 - 6) |
| COVID-19 concern | Mean ± SD | 4.6 ± 1.9 |
| | Median (IQR) | 5 (3 - 6) |
| Have you tested positive for COVID-19? | | 70 (12.4%) |
| Have any of your family members tested positive for COVID-19? | | 517 (74.1%) |

Self-reported health status was measured by the question "How would you describe your current health?" on a 7-point Likert scale (1 = very poor, 7 = excellent). COVID-19 concern was measured on a 7-point Likert scale (1 = not at all concerned, 7 = extremely concerned). AUC, American University in Cairo; BAU, Beirut Arab University; SD, standard deviation; IQR, interquartile range; BMI, body mass index; COPD, chronic obstructive pulmonary disease; CVD, cardiovascular disease.

Less intrusive nudges received the highest levels of approval. Information disclosure (rank 1) was endorsed by 86.5% of participants, and moral appeals such as elderly letters (rank 2) by 84.1%. Similarly, lower-level choice architecture nudges showed strong approval: supermarket floor markers (rank 3, 95.4%), prominent display of fruits and vegetables in grocery apps (rank 4, 88.8%), and park area divisions (rank 5, 82.0%).

By contrast, approval declined for more intrusive measures. Social norm enforcement nudges were less favored: billboard spoilers (rank 6) were approved by 52.0% of participants, and publishing nationalities of curfew violators (rank 7) by only 49.9%. Surveillance-based GPS tracking (rank 8), the most intrusive nudge tested, received intermediate approval at 72.8%.

**Table 2. Approval of COVID-19-related nudges by category and intrusiveness rank (N = 698).**

| Nudge description | Intrusiveness rank | Category | Response | n (%) |
|---|---|---|---|---|
| To increase compliance with social distancing rules, causes of infections are made publicly available (e.g., because of the exchange of hugs and kisses at a family gathering, nine cases of COVID-19 have been detected, three of which required hospitalization). | 1 | Information disclosure | Approve | 604 (86.5) |
| | | | Disapprove | 94 (13.5) |
| To increase compliance with COVID-19 preventive measures, elderly people are asked to send letters to family members pleading with them to respect the rules for their sake. | 2 | Moral nudges | Approve | 587 (84.1) |
| | | | Disapprove | 111 (15.9) |
| To encourage customers to maintain a safe distance of 2 meters between each other, supermarkets are required to install social distancing floor markers at check-out lanes. | 3 | Choice architecture | Approve | 666 (95.4) |
| | | | Disapprove | 32 (4.6) |
| To increase healthy eating during the pandemic, grocery stores are required to display fruits and vegetables as the first items on their mobile apps and web shops. | 4 | Choice architecture | Approve | 620 (88.8) |
| | | | Disapprove | 78 (11.2) |
| To deter people from gathering in large numbers, popular parks and green areas are divided into squares where no more than 10 people can gather in the same square. | 5 | Choice architecture | Approve | 572 (82.0) |
| | | | Disapprove | 126 (18.0) |
| To discourage people from all non-essential road travel during the national lock-down, a campaign is launched advertising spoilers of popular television series on billboards. | 6 | Social norm enforcement | Approve | 363 (52.0) |
| | | | Disapprove | 335 (48.0) |
| To encourage compliance with the national lockdown rules, the number of people who violate the national curfew and their respective nationalities are publicly published. | 7 | Social norm enforcement | Approve | 348 (49.9) |
| | | | Disapprove | 350 (50.1) |
| To limit the spread of the coronavirus, an app uses a GPS feature to track users and gather health information about healthy individuals, and those in quarantine. | 8 | Surveillance-based intervention | Approve | 508 (72.8) |
| | | | Disapprove | 190 (27.2) |

Intrusiveness rank follows the Nuffield Council on Bioethics Intervention Ladder, adapted for this study (S1 Table), where 1 = least intrusive (Provide information) and 8 = most intrusive (Restrict choice).

## Factors associated with nudge approval

We examined whether approval of nudges varied according to age, levels of concern about COVID-19, gender, occupation, family infection status, and income.

Age was found to be a significant factor in nudge approval. As shown in Table 3, older participants (≥22 years) showed higher approval rates for certain nudges compared to younger participants (<22 years). This difference was statistically significant for the spoilers on billboard nudge (56.2% vs. 47.7%, p = 0.024) and for publication of nationalities of curfew violators (54.3% vs. 45.4%, p = 0.019).

COVID-19 concern levels also demonstrated a strong association with nudge approval, as evident in Table 4. Participants with higher COVID-19 concern levels showed significantly higher approval rates for several nudges. For instance, GPS tracking was approved by 78.6% of high-concern participants compared to 66.5% of low-concern participants (p < 0.001). Similarly, the spoilers on billboard nudge was approved by 59.1% of high-concern participants versus 44.3% of low-concern participants (p < 0.001). Approval was also higher among high-concern participants for publication of curfew violators' nationalities (57.4% vs. 41.6%, p < 0.001) and elderly letters (87.6% vs. 80.2%, p = 0.008).

Gender was another significant factor. Females were more likely than males to approve of GPS tracking (76.0% vs. 68.5%, p = 0.027) and supermarket floor markers (97.3% vs. 93.0%, p = 0.007), with full results by gender shown in Table 5.

Occupation was also associated with differences. Faculty and administrative staff showed higher approval than students for billboard spoilers (61.0% vs. 47.3%, p < 0.001) and for publication of curfew violators' nationalities (55.6% vs. 46.8%, p = 0.03). No significant differences were observed for the other nudges. Full results are presented in Table 6.

**Table 3. Approval of COVID-19-related nudges by age.**

| Variables | | Age < 22 | Age >= 22 | p-value |
|---|---|---|---|---|
| To limit the spread of the coronavirus, an app uses a GPS feature to track users and gather health information about healthy individuals, and those in quarantine. | Approve | 244 (70.5%) | 264 (75.0%) | 0.18 |
| | Disapprove | 102 (29.5%) | 88 (25.0%) | |
| To discourage people from all non-essential road travel during the national lockdown, a campaign is launched advertising spoilers of popular television series on billboards. | Approve | 165 (47.7%) | 198 (56.2%) | **0.024** |
| | Disapprove | 181 (52.3%) | 154 (43.8%) | |
| To encourage customers to maintain a safe distance of 2 meters between each other, supermarkets are required to install social distancing floor markers at checkout lanes. | Approve | 327 (94.5%) | 339 (96.3%) | 0.26 |
| | Disapprove | 19 (5.5%) | 13 (3.7%) | |
| To encourage compliance with the national lockdown rules, the number of people who violate the national curfew and their respective nationalities are publicly published. | Approve | 157 (45.4%) | 191 (54.3%) | **0.019** |
| | Disapprove | 189 (54.6%) | 161 (45.7%) | |
| To deter people from gathering in large numbers, popular parks and green areas are divided into squares where no more than 10 people can gather in the same square. | Approve | 277 (80.1%) | 295 (83.8%) | 0.20 |
| | Disapprove | 69 (19.9%) | 57 (16.2%) | |
| To increase compliance with social distancing rules, causes of infections are made publicly available (e.g., because of the exchange of hugs and kisses at a family gathering, nine cases of COVID-19 have been detected of which three are being hospitalized). | Approve | 298 (86.1%) | 306 (86.9%) | 0.76 |
| | Disapprove | 48 (13.9%) | 46 (13.1%) | |
| To increase compliance with COVID-19 preventive measures, elderly people are asked to send letters to family members pleading with them to respect the rules for their sake. | Approve | 287 (83.0%) | 300 (85.2%) | 0.41 |
| | Disapprove | 59 (17.0%) | 52 (14.8%) | |
| To increase healthy eating during the pandemic, grocery stores are required to display fruits and vegetables as the first items on their mobile apps and web shops. | Approve | 306 (88.4%) | 314 (89.2%) | 0.75 |
| | Disapprove | 40 (11.6%) | 38 (10.8%) | |

GPS, global positioning system.

**Table 4. Approval of COVID-19-related nudges by COVID-19 concern level.**

| Variables | | COVID-19 concern < 5 | COVID-19 concern >= 5 | p-value |
|---|---|---|---|---|
| To limit the spread of the coronavirus, an app uses a GPS feature to track users and gather health information about healthy individuals, and those in quarantine. | Approve | 222 (66.5%) | 286 (78.6%) | **<0.001** |
| | Disapprove | 112 (33.5%) | 78 (21.4%) | |
| To discourage people from all non-essential road travel during the national lockdown, a campaign is launched advertising spoilers of popular television series on billboards. | Approve | 148 (44.3%) | 215 (59.1%) | **<0.001** |
| | Disapprove | 186 (55.7%) | 149 (40.9%) | |
| To encourage customers to maintain a safe distance of 2 meters between each other, supermarkets are required to install social distancing floor markers at checkout lanes. | Approve | 314 (94.0%) | 352 (96.7%) | 0.089 |
| | Disapprove | 20 (6.0%) | 12 (3.3%) | |
| To encourage compliance with the national lockdown rules, the number of people who violate the national curfew and their respective nationalities are publicly published. | Approve | 139 (41.6%) | 209 (57.4%) | **<0.001** |
| | Disapprove | 195 (58.4%) | 155 (42.6%) | |
| To deter people from gathering in large numbers, popular parks and green areas are divided into squares where no more than 10 people can gather in the same square. | Approve | 264 (79.0%) | 308 (84.6%) | 0.056 |
| | Disapprove | 70 (21.0%) | 56 (15.4%) | |
| To increase compliance with social distancing rules, causes of infections are made publicly available (e.g., because of the exchange of hugs and kisses at a family gathering, nine cases of COVID-19 have been detected of which three are being hospitalized). | Approve | 284 (85.0%) | 320 (87.9%) | 0.27 |
| | Disapprove | 50 (15.0%) | 44 (12.1%) | |
| To increase compliance with COVID-19 preventive measures, elderly people are asked to send letters to family members pleading with them to respect the rules for their sake. | Approve | 268 (80.2%) | 319 (87.6%) | **0.008** |
| | Disapprove | 66 (19.8%) | 45 (12.4%) | |
| To increase healthy eating during the pandemic, grocery stores are required to display fruits and vegetables as the first items on their mobile apps and web shops. | Approve | 295 (88.3%) | 325 (89.3%) | 0.69 |
| | Disapprove | 39 (11.7%) | 39 (10.7%) | |

GPS, global positioning system.

**Table 5. Approval of COVID-19-related nudges by gender.**

| Variables | | Males (n=298, 42.7%) | Females (n=400, 57.3%) | p-value |
|---|---|---|---|---|
| To limit the spread of the coronavirus, an app uses a GPS feature to track users and gather health information about healthy individuals, and those in quarantine. | Approve | 204 (68.46%) | 304 (76%) | **0.027** |
| | Disapprove | 94 (31.54%) | 96 (24%) | |
| To discourage people from all non-essential road travel during the national lockdown, a campaign is launched advertising spoilers of popular television series on billboards. | Approve | 158 (53.02%) | 205 (51.25%) | 0.643 |
| | Disapprove | 140 (46.98%) | 195 (48.75%) | |
| To encourage customers to maintain a safe distance of 2 meters between each other, supermarkets are required to install social distancing floor markers at checkout lanes. | Approve | 277 (92.95%) | 389 (97.25%) | **0.007** |
| | Disapprove | 21 (7.05%) | 11 (2.75%) | |
| To encourage compliance with the national lockdown rules, the number of people who violate the national curfew and their respective nationalities are publicly published. | Approve | 152 (51.01%) | 196 (49%) | 0.6 |
| | Disapprove | 146 (48.99%) | 204 (51%) | |
| To deter people from gathering in large numbers, popular parks and green areas are divided into squares where no more than 10 people can gather in the same square. | Approve | 239 (80.2%) | 333 (83.25%) | 0.3 |
| | Disapprove | 59 (19.8%) | 67 (16.75%) | |
| To increase compliance with social distancing rules, causes of infections are made publicly available (e.g., because of the exchange of hugs and kisses at a family gathering, nine cases of COVID-19 have been detected of which three are being hospitalized). | Approve | 254 (85.23%) | 350 (87.5%) | 0.386 |
| | Disapprove | 44 (14.77%) | 50 (12.5%) | |
| To increase compliance with COVID-19 preventive measures, elderly people are asked to send letters to family members pleading with them to respect the rules for their sake. | Approve | 255 (85.57%) | 332 (83%) | 0.358 |
| | Disapprove | 43 (14.43%) | 68 (17%) | |
| To increase healthy eating during the pandemic, grocery stores are required to display fruits and vegetables as the first items on their mobile apps and web shops. | Approve | 260 (87.25%) | 360 (90%) | 0.254 |
| | Disapprove | 38 (12.75%) | 40 (10%) | |

Participants with a family member who had tested positive for COVID-19 were more likely to approve of GPS tracking (79.6% vs. 70.4%, p=0.017), but no other nudges differed significantly by this factor. These results are presented in S2 Table. Analysis by income level did not show significant differences in approval across low-, middle-, and high-income groups (S3 Table).

## Discussion

This study provides novel insights into the perception and approval of COVID-19-related nudges across four Arab countries in a university setting. Our findings reveal a generally high level of approval for nudges, ranging from 49.9% to 95.4%. However, we observed significant variations in approval rates depending on the type of nudge and sociodemographic factors, and these results should not be assumed to represent the general population.

In this study, we selected eight nudges commonly used in behavioral economics literature to assess public perceptions and acceptability of nudge-based COVID-19-related interventions. These included: choice architecture nudges, such as floor markers in supermarkets and the prominent placement of fruits and vegetables in grocery apps; information disclosure nudges, such as publicly sharing the causes of COVID-19 infections; moral appeals, including letters from elderly individuals urging family members to follow lockdown measures; social norm enforcement nudges, including public shaming of lockdown violators and the use of spoilers on billboards to deter non-essential travel; and surveillance-based interventions, such as GPS tracking of individuals in quarantine. These categories reflect distinct behavioral mechanisms through which nudges are intended to influence individual decision-making during a public health crisis.

**Table 6. Approval of COVID-19-related nudges by occupation (students vs. staff).**

| Variables | | Student n=457 | Others (Academic Staff/Admin/ Faculty) n=241 | p-value |
|---|---|---|---|---|
| To limit the spread of the coronavirus, an app uses a GPS feature to track users and gather health information about healthy individuals, and those in quarantine. | Approve | 332 (72.6%) | 176 (73.0%) | 0.93 |
| | Disapprove | 125 (27.4%) | 65 (27.0%) | |
| To discourage people from all non-essential road travel during the national lockdown, a campaign is launched advertising spoilers of popular television series on billboards. | Approve | 216 (47.3%) | 147 (61.0%) | **<0.001** |
| | Disapprove | 241 (52.7%) | 94 (39.0%) | |
| To encourage customers to maintain a safe distance of 2 meters between each other, supermarkets are required to install social distancing floor markers at checkout lanes. | Approve | 433 (94.7%) | 233 (96.7%) | 0.25 |
| | Disapprove | 24 (5.3%) | 8 (3.3%) | |
| To encourage compliance with the national lockdown rules, the number of people who violate the national curfew and their respective nationalities are publicly published | Approve | 214 (46.8%) | 134 (55.6%) | **0.03** |
| | Disapprove | 243 (53.2%) | 107 (44.4%) | |
| To deter people from gathering in large numbers, popular parks and green areas are divided into squares where no more than 10 people can gather in the same square. | Approve | 371 (81.2%) | 201 (83.4%) | 0.47 |
| | Disapprove | 86 (18.8%) | 40 (16.6%) | |
| To increase compliance with social distancing rules, causes of infections are made publicly available (e.g., because of the exchange of hugs and kisses at a family gathering, nine cases of COVID-19 have been detected of which three are being hospitalized). | Approve | 393 (86.0%) | 211 (87.6%) | 0.57 |
| | Disapprove | 64 (14.0%) | 30 (12.4%) | |
| To increase compliance with COVID-19 preventive measures, elderly people are asked to send letters to family members pleading with them to respect the rules for their sake. | Approve | 379 (82.9%) | 208 (86.3%) | 0.25 |
| | Disapprove | 78 (17.1%) | 33 (13.7%) | |
| To increase healthy eating during the pandemic, grocery stores are required to display fruits and vegetables as the first items on their mobile apps and web shops. | Approve | 408 (89.3%) | 212 (88.0%) | 0.60 |
| | Disapprove | 49 (10.7%) | 29 (12.0%) | |

These nudges are widely tested in Western contexts, but their applicability in non-Western settings is less understood. Given the distinct cultural characteristics of Arab societies, this study sought to examine how these nudges are perceived and received in a different cultural environment.

Finally, the types of nudges selected also align well with Hofstede's cultural dimensions. According to Hofstede's country comparison tool [17], Arab societies (specifically, Egypt, Lebanon, UAE, and KSA) exhibit a high average power distance index of 72, indicating a strong acceptance of hierarchical authority and unequal power distribution. This cultural trait suggests that social norm enforcement and surveillance-based nudges may be effective, as directives from trusted authorities are more likely to be followed without resistance. Additionally, the relatively high uncertainty avoidance index of 61 [17], reflects a preference for clear structures and rules to minimize ambiguity. Accordingly, choice architecture nudges, such as floor markers, and information disclosure nudges, such as infection cause reporting, provide explicit guidance and reduce uncertainty, promoting compliance with health protocols. Moral appeals, such as letters from elderly family members urging adherence to guidelines, were selected in light of the region's low individualism index (average score of 31) [17], reflecting strong familial ties and collective responsibility. Overall, the selection of nudges was guided by behavioral science literature, expert recommendations, real-time policy relevance, and their cultural and contextual resonance within the Arab region.

Our results indicate that approval of nudges varied systematically by intrusiveness level. Less intrusive interventions received the strongest endorsement: information disclosure (rank 1), moral appeals (rank 2), and lower-level choice architecture nudges (ranks 3–5) all showed approval rates above 80%, with supermarket floor markers (rank 3) reaching the highest at 95.4%. In contrast, more intrusive nudges attracted less support. Social norm enforcement

interventions—billboard spoilers (rank 6, 52.0%) and publishing violators' nationalities (rank 7, 49.9%)—were the least favored. The most intrusive measure tested, GPS tracking (rank 8), received intermediate approval (72.8%).

When interpreting the results by nudge category, our results indicate that choice architecture and information disclosure nudges received the highest approval rates, while social norm enforcement nudges were less favored. This aligns with previous research by Sunstein et al. that found a general preference for nudges that are transparent and preserve freedom of choice [18]. The high approval rates for floor markers (95.4%) and prominent display of fruits and vegetables (88.8%) suggest that Arabs are receptive to nudges that provide guidance without limiting personal autonomy.

Interestingly, the moral nudge involving elderly sending letters received high approval (84.1%), which may reflect the strong family ties and respect for elders in Arab culture [9]. This finding extends our understanding of culturally specific nudges and their potential effectiveness in non-Western contexts.

The lower approval rates for social norm enforcement nudges, particularly public shaming (49.9%), align with findings from Jung and Mellers' study in the United States, suggesting a cross-cultural aversion to nudges that may be perceived as infringing on personal privacy or dignity [13]. Our findings on the relatively high approval of surveillance-based interventions nudges, such as GPS tracking (72.8%), contrast with previous studies conducted in Western contexts, where similar interventions have typically received lower acceptance due to heightened privacy concerns [18,19]. This divergence likely reflects deeper cultural differences in societal attitudes toward individual autonomy, privacy, and governmental authority. Arab societies, often characterized by collectivist values and stronger emphasis on community responsibility, might prioritize public health and collective safety over individual autonomy during crises [8]. Additionally, variations in trust towards government and public health authorities between Arab and Western contexts may further explain the greater acceptance of measures like GPS tracking observed in our study [20]. These results align with cultural theories such as Hofstede's dimensions [8], highlighting the critical importance of cultural context when designing and implementing behavioral interventions.

Our study revealed significant associations between nudge approval and various sociodemographic factors. The positive correlation between age and approval rates for certain nudges, particularly the spoilers on billboard nudge, aligns with findings in other studies [13,19,21]. This age effect may be attributed to increased risk perception among older individuals during the pandemic [22].

The higher approval rates among female participants for nudges such as GPS tracking and floor markers corroborate the findings of Ölander and Thøgersen, who reported gender differences in receptiveness to pro-environmental nudges [23]. This gender effect may be related to differences in risk perception or social responsibility attitudes between men and women [24].

The strong positive association between COVID-19 concern levels and nudge approval across all nudge types is a key finding of our study. This relationship underscores the importance of risk perception in shaping attitudes towards behavioral interventions, as highlighted by Slovic in his seminal work on risk perception [25]. Our results suggest that public health communication strategies that effectively convey the seriousness of health risks may increase receptiveness to nudges [8]. The strong influence of COVID-19 concern levels on nudge approval further underscores the importance of effective, culturally appropriate risk communication in public health strategies.

Our findings offer preliminary insights that may be relevant to policymakers and public health officials in the Arab world and beyond. The high approval rates for choice architecture and information disclosure nudges suggest that these types of interventions may be more acceptable and well-received in Arab countries. However, the lower approval for social norm enforcement nudges highlights the need for careful consideration of privacy concerns and cultural sensitivities when designing interventions.

While these findings provide preliminary insights, further research is required to evaluate whether such nudges are effective in practice. Policymakers may consider these findings as a basis for exploring culturally appropriate behavioral

interventions, but implementation should be directed by evidence from future real-world studies, such as observational or experimental studies.

While our study provides valuable insights, it has several limitations. This study, while consistent with expectations for voluntary email-based surveys, is limited by potential selection bias due to a low response rate and a sample that may not be representative of the broader target population. The sample was predominantly composed of university students and staff, which may not fully represent the broader Arab population, particularly older adults, rural communities, and lower-income groups. To partially address this limitation, we conducted subgroup analyses based on key demographic and contextual variables, including age, gender, occupation, degree of concern about COVID-19, income, and family COVID-19 status. However, these subgroup analyses remain limited by the fact that the data were drawn exclusively from a university setting and may not reflect the perspectives of the general population. The cross-sectional nature of our study limits our ability to assess how attitudes towards nudges may have evolved over the course of the pandemic. Longitudinal studies could provide valuable insights into the dynamics of nudge perception over time. Additionally, the study relied on hypothetical nudges rather than actual behavioral responses in real-world settings, which limits the extent to which conclusions about effectiveness can be drawn. Future research should involve longitudinal and field studies with more demographically diverse participants. Moreover, other important factors such as trust in government, exposure to misinformation, and religious beliefs—which may strongly influence attitudes toward nudges—were not measured and should be considered in future research.

## Conclusion

This study contributes to the growing body of literature on cross-cultural perceptions of nudges, providing novel insights into the Arab context. Our findings reveal a generally positive attitude towards COVID-19-related nudges in the studied Arab countries, with important variations based on nudge type and sociodemographic factors. These findings reflect the perspectives of university affiliates and may not be generalizable to the broader population. These results underscore the potential of nudging as a public health strategy in the Arab world, while also highlighting the need for culturally sensitive and context-specific approaches to behavioral interventions. As the global community continues to grapple with public health challenges, understanding these cultural nuances in nudge perception becomes increasingly crucial for effective policy-making and public health management.

## Supporting information

**S1 Checklist.  Inclusivity in global research checklist.**
(DOCX)

**S1 Table.  Mapping of interventions onto the Nuffield Council on Bioethics Intervention Ladder.**
(DOCX)

**S2 Table.  Approval of COVID-19-related nudges by family COVID-19 status.**
(DOCX)

**S3 Table.  Approval of COVID-19-related nudges by income.**
(DOCX)

## Author contributions

**Conceptualization:** Fadi Makki, Cass R. Sunstein.

**Data curation:** Belal Nedal Sabbah, Mariam Abdelnabi, Paola Schietekat, Nabil Saleh, Abdulkarim Almakadma, Mohamed Al-Komi, Ebaa Alsayed, Rajaa Fakhoury, Fatima Saleh, Basema Saddik, Hana Fakhoury.

**Formal analysis:** Hani Tamim.

**Investigation:** Fadi Makki, Belal Nedal Sabbah, Mariam Abdelnabi, Paola Schietekat, Nabil Saleh, Ali Osseiran, Abdulkarim Almakadma, Mohamed Al-Komi, Ebaa Alsayed, Rajaa Fakhoury, Fatima Saleh, Basema Saddik, Hana Fakhoury.

**Methodology:** Fadi Makki, Hani Tamim, Mariam Abdelnabi, Paola Schietekat, Nabil Saleh, Ali Osseiran.

**Project administration:** Fadi Makki, Abdulkarim Almakadma, Mohamed Al-Komi, Ebaa Alsayed, Rajaa Fakhoury, Fatima Saleh, Basema Saddik, Hana Fakhoury, Sarah Daher.

**Supervision:** Fadi Makki, Cass R. Sunstein.

**Validation:** Hani Tamim.

**Writing – original draft:** Belal Nedal Sabbah.

**Writing – review & editing:** Fadi Makki, Belal Nedal Sabbah, Hani Tamim, Mariam Abdelnabi, Paola Schietekat, Nabil Saleh, Ali Osseiran, Abdulkarim Almakadma, Mohamed Al-Komi, Ebaa Alsayed, Rajaa Fakhoury, Fatima Saleh, Basema Saddik, Hana Fakhoury, Sarah Daher, Cass R. Sunstein.

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
