## [Decision Letter · Decision Letter 0]

31 Mar 2025

PGPH-D-25-00340

Perceptions of COVID-19-related nudges in the Arab world: A cross-country analysis of approval rates and associated factors

Dear Dr. Fakhoury,

Thank you for submitting your manuscript to PLOS Global Public Health. After careful consideration, we feel that it has merit but does not fully meet PLOS Global Public Health’s publication criteria as it currently stands. Therefore, we invite you to submit a revised version of the manuscript that addresses the points raised during the review process.

Please refer to reviewer's comments and address all of them. There are no contradictory comments in the reviewer's suggestions that should cause confusion in the revision of the manuscript. Pay particular attention to PLOS Global Public Health policy regarding data availability.

In addition to reviewers' comments, authors should refrain from conclusions or interpretation not sustained by the study. For instance, since the effectivity of the actual nudges was not evaluated, it can hardly be advisable for policy-makers to implement a nudge strategy based only on the results from this study.

We look forward to receiving your revised manuscript.

Kind regards,

Miguel Reina Ortiz, M.D., M.S., M.P.H., M.P.T., Ph.D.

Academic Editor

Journal Requirements:

1. Please include a complete copy of PLOS’ questionnaire on inclusivity in global research in your revised manuscript. Our policy for research in this area aims to improve transparency in the reporting of research performed outside of researchers’ own country or community. The policy applies to researchers who have travelled to a different country to conduct research, research with Indigenous populations or their lands, and research on cultural artefacts. The questionnaire can also be requested at the journal’s discretion for any other submissions, even if these conditions are not met. Please find more information on the policy and a link to download a blank copy of the questionnaire here: https://journals.plos.org/globalpublichealth/s/best-practices-in-research-reporting. Please upload a completed version of your questionnaire as Supporting Information when you resubmit your manuscript. 2. In the online submission form, you indicated that The data sets used and/or analyzed during the current study are available from the authors upon reasonable request.  All PLOS journals now require all data underlying the findings described in their manuscript to be freely available to other researchers, either 1. In a public repository, 2. Within the manuscript itself, or 3. Uploaded as supplementary information. This policy applies to all data except where public deposition would breach compliance with the protocol approved by your research ethics board. If your data cannot be made publicly available for ethical or legal reasons (e.g., public availability would compromise patient privacy), please explain your reasons by return email and your exemption request will be escalated to the editor for approval. Your exemption request will be handled independently and will not hold up the peer review process, but will need to be resolved should your manuscript be accepted for publication. One of the Editorial team will then be in touch if there are any issues.

Additional Editor Comments (if provided):

Thank you very much for your submission. Please refer to reviewer's comments and address all of them. There are no contradictory comments in the reviewer's suggestions that should cause confusion in the revision of the manuscript. Pay particular attention to PLOS Global Public Health policy regarding data availability.

In addition to reviewers' comments, authors should refrain from conclusions or interpretation not sustained by the study. For instance, since the effectivity of the actual nudges was not evaluated, it can hardly be advisable for policy-makers to implement a nudge strategy based only on the results from this study.

Reviewers' comments:

Reviewer's Responses to Questions

**Comments to the Author**

1. Does this manuscript meet PLOS Global Public Health’s publication criteria ? Is the manuscript technically sound, and do the data support the conclusions? The manuscript must describe methodologically and ethically rigorous research with conclusions that are appropriately drawn based on the data presented.

Reviewer #1: Yes

Reviewer #2: Yes

2. Has the statistical analysis been performed appropriately and rigorously?

Reviewer #1: Yes

Reviewer #2: Yes

3. Have the authors made all data underlying the findings in their manuscript fully available (please refer to the Data Availability Statement at the start of the manuscript PDF file)?

Reviewer #1: Yes

Reviewer #2: No

4. Is the manuscript presented in an intelligible fashion and written in standard English?

Reviewer #1: Yes

Reviewer #2: Yes

5. Review Comments to the Author

Reviewer #1: Thank you for the opportunity to review this manuscript. The study on COVID-19-related nudges is interesting, given the severity of the crisis and the cultural context of the Arab world. Here are my comments, which I hope will be useful for your revision.

First, I believe that the inclusion of data from the four Arab countries (Egypt, Lebanon, Saudi Arabia, and the UAE) strengthens the generalizability of the findings within the region. That said, because studies on nudges have generally come from the West, and you employ methodologies from such studies, you should enhance the discussion on its comparisons to the existing literature, which you only do so in brief (e.g., Sunstein et al.). In particular, expand the discussion on why certain nudges (e.g., GPS tracking) are more accepted in the Arab world than in Western contexts, possibly due to differing attitudes toward privacy and public health priorities. This discussion will highlight the importance of understanding variations in public perceptions based on cultural and national differences when developing nudge interventions.

Your theory development could be stronger. You used Hofstede’s cultural dimensions but do not connect those dimensions to the nudges. Please discuss how specific cultural dimensions (e.g., power distance, uncertainty avoidance, etc.) explains the differences in the acceptance of intrusive versus instructive nudges. Also, please note that Hofstede's dimensions have been criticized for its methodology and generalizability (Venaik, S. and Brewer, P. (2013), "Critical issues in the Hofstede and GLOBE national culture models", International Marketing Review, Vol. 30 No. 5, pp. 469-482. https://doi.org/10.1108/IMR-03-2013-0058). You may want to address this in your methods and say why the framework is appropriate for this study.

The sample consists primarily of university students and academic staff, which may not fully represent broader demographic groups in the Arab world. At minimum, you should acknowledge this limitation and discuss how future studies could include more diverse participants (e.g., older adults, rural populations, lower-income groups). You can also try to address this limitation with sub-group analyses by the demographic variables in your data to determine if there are systematic biases.

I do not understand how you categorized the nudges into intrusive, instructive, informative, entertainment, and moral. Please discuss how you derived these categories with a stronger theoretical or empirical basis for the classifications, possibly referencing the literature.

Other variables, such as trust in government, and exposure to misinformation, or religious beliefs, may influence nudge approval but are not examined. These are potential confounders and should be addressed in how future research should explore their role in shaping public attitudes toward nudges.

My biggest concern with this study is that it evaluates hypothetical nudges rather than real-world interventions. Hence, there is a limit to what we can learn from this paper. You might want to explore further how perceptions of nudges might differ if implemented in real settings and suggest avenues for experimental or observational studies. This weakness makes the policy discussion less meaningful. While you mention the importance of considering cultural sensitivities, you can't provide specific recommendations for designing and implementing nudges in Arab countries.

Reviewer #2: This study examines public approval of COVID-19-related nudges in Egypt, Lebanon, Saudi Arabia, and the UAE. Using survey data from 698 participants, it finds high overall support (76%), especially for informative and instructive nudges. Intrusive nudges were less favored. Approval varied by country, demographics, and level of COVID-19 concern. The findings suggest nudging can be effective in the Arab world if culturally tailored.

This is an interesting paper that presents novel findings from a region where public opinion data—particularly on behavioral interventions like nudges—is relatively scarce. The cross-country design and focus on COVID-19-related nudges in the Arab world are valuable contributions. The results are clear and relevant for both policy and behavioral science. That said, the paper would benefit from addressing several issues outlined below.

Comments:

1. In the introduction, the authors state that “the effectiveness of nudging has been well-documented in Western contexts.” This phrasing may imply that nudges are always effective, which is not the case. I recommend rephrasing to acknowledge the mixed evidence and contextual variability in nudge effectiveness.

2. The Materials and Methods section does not explain how participants were recruited or how the survey was distributed. Was it via flyers, social media, email lists, or another method? These details are important for assessing sample representativeness and potential biases.

3. The authors state that informed consent was obtained while maintaining anonymity. More detail is needed: how was consent collected in practice (e.g., online checkbox, written form)? How was anonymity preserved during this process?

4. The approval/disapproval measure is binary. The authors should briefly explain why this approach was chosen over a Likert-type or continuous scale, which could capture more nuanced attitudes.

5. Some categorizations are unclear. For example, the prominent display of fruits and vegetables is labeled as “informative,” though it may be more accurately classified as environmental or design-based. The “spoilers” nudge is categorized as “entertainment,” but that label may not effectively convey its behavioral mechanism. Please clarify the reasoning behind these classifications.

6. There should be more discussion about how the eight nudges were selected. For instance, the fruits and vegetables display is not COVID-related—why was it included? Were any of the nudges based on actual policies implemented in the studied countries? If so, noting this would strengthen the paper.

7. It would be helpful to report the distribution of the total number of nudges approved by respondents, both overall and by country. This aggregate measure could also serve as a useful dependent variable in additional analyses.

6. PLOS authors have the option to publish the peer review history of their article (what does this mean? ). If published, this will include your full peer review and any attached files.

**Do you want your identity to be public for this peer review?** For information about this choice, including consent withdrawal, please see our Privacy Policy .

Reviewer #1: No

Reviewer #2: No

---

## [Decision Letter · Decision Letter 1]

10 Jun 2025

PGPH-D-25-00340R1

Perceptions of COVID-19-related nudges in the Arab world: A cross-country analysis of approval rates and associated factors

Dear Dr. Fakhoury,

Thank you for submitting your manuscript to PLOS Global Public Health. After careful consideration, we feel that it has merit but does not fully meet PLOS Global Public Health’s publication criteria as it currently stands. Therefore, we invite you to submit a revised version of the manuscript that addresses the points raised during the review process.

We look forward to receiving your revised manuscript.

Kind regards,

Miguel Reina Ortiz, M.D., M.S., M.P.H., M.P.T., Ph.D.

Academic Editor

Additional Editor Comments (if provided):

Reviewers' comments:

Reviewer's Responses to Questions

**Comments to the Author**

1. If the authors have adequately addressed your comments raised in a previous round of review and you feel that this manuscript is now acceptable for publication, you may indicate that here to bypass the “Comments to the Author” section, enter your conflict of interest statement in the “Confidential to Editor” section, and submit your "Accept" recommendation.

Reviewer #1: All comments have been addressed

Reviewer #2: (No Response)

2. Does this manuscript meet PLOS Global Public Health’s publication criteria ? Is the manuscript technically sound, and do the data support the conclusions? The manuscript must describe methodologically and ethically rigorous research with conclusions that are appropriately drawn based on the data presented.

Reviewer #1: Yes

Reviewer #2: Partly

3. Has the statistical analysis been performed appropriately and rigorously?

Reviewer #1: Yes

Reviewer #2: Yes

4. Have the authors made all data underlying the findings in their manuscript fully available (please refer to the Data Availability Statement at the start of the manuscript PDF file)?

Reviewer #1: Yes

Reviewer #2: (No Response)

5. Is the manuscript presented in an intelligible fashion and written in standard English?

Reviewer #1: Yes

Reviewer #2: (No Response)

6. Review Comments to the Author

Reviewer #1: No further comments. My concerns have been addressed

Reviewer #2: The authors have revised the manuscript in response to the initial review, and several of the comments have been satisfactorily addressed. However, some important concerns remain unresolved or only partially addressed. Below is a point-by-point assessment of the extent to which each of the original comments has been addressed:

1. Claim about the effectiveness of nudging in Western contexts

Addressed. The revised phrasing in the introduction appropriately reflects the mixed evidence and contextual variability in the effectiveness of nudging. This change improves the balance and nuance of the opening section.

2. Participant recruitment and sample representativeness

Partly addressed. While the authors now mention that recruitment was conducted via institutional mailing lists, they do not report how many emails were sent, what the response rate was, or whether the sample is representative of the broader target population. This information is important for assessing potential selection bias and the generalizability of the results. I recommend including these additional details.

3. Consent and anonymity procedures

Addressed. The manuscript now clearly explains how informed consent was obtained and how anonymity was preserved in the online survey context.

4. Binary approval measure

Addressed. The authors have provided a reasonable justification for using a binary measure.

5. Unclear categorization of nudges

Not addressed. Several categorizations remain unclear or inadequately justified. For example, the prominent display of fruits and vegetables continues to be labeled as “informative,” whereas this nudge is better understood as a salience-based or choice architecture intervention aimed at influencing defaults or attention. Similarly, the classification of the TV spoiler intervention as “entertainment” fails to capture its behavioral mechanism—namely, deterrence through aversive social sanctioning.

I strongly encourage the authors to revise the typology of nudges using more analytically grounded behavioral categories, such as:

-Choice architecture / salience

-Information disclosure / feedback

-Emotional or moral appeals

-Social norm enforcement / sanctions

-Coercive or surveillance-based interventions

Reclassifying nudges in this way (or in a similar way) would improve conceptual clarity and strengthen alignment with established behavioral economics frameworks.

6. Selection and justification of nudges

Not addressed. The rationale for including the eight specific nudges remains unclear. While the authors reference Hofstede’s cultural dimensions, they do not explain how these dimensions guided the selection of nudges.

Regarding the healthy eating nudge (display of fruits and vegetables), which is not specific to COVID-19, the authors now state that healthy eating was promoted during the pandemic to improve immunity in the studied countries, but they do not provide citations or supporting evidence. A reference or link to relevant policy guidance or public communications would be helpful to support this claim.

7. Distribution of total nudges approved

Addressed.

7. PLOS authors have the option to publish the peer review history of their article (what does this mean? ). If published, this will include your full peer review and any attached files.

**Do you want your identity to be public for this peer review?** For information about this choice, including consent withdrawal, please see our Privacy Policy .

Reviewer #1: No

Reviewer #2: No

---

## [Decision Letter · Decision Letter 2]

14 Aug 2025

PGPH-D-25-00340R2

Perceptions of COVID-19-related nudges in the Arab world: A cross-country analysis of approval rates and associated factors

Dear Dr. Fakhoury,

Thank you for submitting your manuscript to PLOS Global Public Health. After careful consideration, we feel that it has merit but does not fully meet PLOS Global Public Health’s publication criteria as it currently stands. Therefore, we invite you to submit a revised version of the manuscript that addresses the points raised during the review process.

We look forward to receiving your revised manuscript.

Kind regards,

Miguel Reina Ortiz, M.D., M.S., M.P.H., M.P.T., Ph.D.

Academic Editor

Journal Requirements:

Additional Editor Comments:

Thank you very much for addressing previous comments. Please address the following comments.

1. In the methods and/or conclusions of the abstract, please indicate that the findings apply to university settings and not the general public.

2. Specify what stringent measures where used to maintain participant anonymity and data confidentiality (pages 123-124).

3. Line 127. Specify which Supplemental Information is being references (currently it reads "SX").

4. Specify in methods/results whether data on university affiliation (i.e., student, faculty, staff) was collected; if so, whether it was analyzed; and, if so, whether it had any impact on outcomes.

5. Denote sample size with "n" and universe size with "N."

6. Lines 162-165 state that country (Lebanon, UAE< Saudi Arabia or Egypt) are denoted in Table 1, but Table 1 does not show that information. Please update Table 1.

7. Lines 165-166, assure consistence between what is reported in the text and what is reported in the table. Suggest to report only the most appropriate measures of central tendency and dispersion (i.e., mean and SD OR median and IQR) as opposed to both. Same for number of children in Table 1.

8. In Table 1, report BMI only once (i.e., either as categories OR as "BMI = Healthy").

9. Table 1, add "self-reported" where applicable.

10. Explain definition, measurement and meaning of the following variable: "how would you describe your current health"?

11. In Table 1, report "self-health" and "COVID concern" only once.

12. Line 180, suggest to delete "the" before "choice."

13. In the methods section, "prominent placement of fruits and vegetables" is listed under the "choice architecture" category (Lines 140-141) ) whereas it is listed under the "information disclosure" category (Lines 182-184). Ensure that it is listed under the right category in both occasions.

14. Line 184, suggest to delete "the" before "moral." Also, suggest to write "Moral nudges" as opposed to "Moral nudge."

15. Need to justify how "prominent placement of fruits and vegetables" is a "COVID-19-related nudge."

16. Explain what is meant by "use of spoilers on billboards."

17. Explain under what category of nudges does "use of spoilers on billboards" fall into?

18. Table 2, ensure formatting is correct.

19. Explain in text what "park area divisions" mean.

20. Table 2, identify which category each nudge belong to.

21. Table 2, if not verbatim and explicitly described in the text of the nudge description, identify in parentheses, what type of nudge it is (i.e., public shaming, etc.).

22. Line 198, p<0.05 also found for the "nationalities being published" nudge.

23. Explain how "COVID concern" was measured. What instrument was used? What do the scores mean?

24. Line 204, there were two additional nudges showing p<0.05 associations with "COVID concern."

25. Lines 205-209, add percentages of approval. Also, add table showing this data.

26. Report associations with other variables, for instance, faculty vs. student, income level, medicine/health vs. other (do you have information about whether participants are affiliated with an economics and/or behavioral science department - if so, analyze and report associations), etc..

26. Lines 222-225, specify that this data comes from a university setting, not the general population.

27. Was being faculty vs. student have an impact on COVID?

28. Line 227, do you mean "acceptability" instead of "acceptance". Also, suggest to add "nudge-based" before intervention.

29. Lines 229-235 can be deleted, offers no new insight.

30. Line 238, consider deleting "established."

31. Line 243, clarify whether those (i.e., GPS tracking, social distancing floor markers, and public disclosure of infection causes) are examples of "observed interventions," "illustrative examples," or "actual implementation." Also, how do you distinguish between "observed interventions" and "actual implementation."

32. 249, this aim is different than the one stated at the beginning of the paper (i.e., nudges vs. policies). Be consistent.

33. Lines 240-252, this process of selecting nudges should be moved to the methods section. Also, explain on more detail how experts where consulted and how they provided feedback. Was there a particular methodology used?

34. Explain, in the methods, how did the Hofstede's scale informed nudge selection. Despite being "high power distance" some of the nudges selected range at different levels of power balance/hierarchy/intrusiveness. Did Hofstede's scale play a role in nudge selection? in the analysis? or only in the discussion?

35. Lines 266-270, please move to methods section where nudges and their selection are being explained.

36. Line 304 needs rephrasing, check for grammar and syntax.

37. Lines 319-320, move up, next to 312.

38. Line 329, defined the target population, put that definition in the introduction and/or methods.

39. Line 333-334, this sub-category analysis is still limited by the sample being taken from a university setting, it does not address the limitation.

40. Line 349-351, indicate that the population studied was in a university setting.

GENERAL: not clear if the "public disclosure" nudges disclose aggregated data OR individual, identifiable data with, say, cause of infection or nationality. Please clarify this in the methods section.

Reviewers' comments:

Reviewer's Responses to Questions

**Comments to the Author**

1. If the authors have adequately addressed your comments raised in a previous round of review and you feel that this manuscript is now acceptable for publication, you may indicate that here to bypass the “Comments to the Author” section, enter your conflict of interest statement in the “Confidential to Editor” section, and submit your "Accept" recommendation.

Reviewer #2: All comments have been addressed

2. Does this manuscript meet PLOS Global Public Health’s publication criteria ? Is the manuscript technically sound, and do the data support the conclusions? The manuscript must describe methodologically and ethically rigorous research with conclusions that are appropriately drawn based on the data presented.

Reviewer #2: Yes

3. Has the statistical analysis been performed appropriately and rigorously?

Reviewer #2: Yes

4. Have the authors made all data underlying the findings in their manuscript fully available (please refer to the Data Availability Statement at the start of the manuscript PDF file)?

Reviewer #2: Yes

5. Is the manuscript presented in an intelligible fashion and written in standard English?

Reviewer #2: Yes

6. Review Comments to the Author

Reviewer #2: The issues I had raised have been addressed. I have no more comments.

7. PLOS authors have the option to publish the peer review history of their article (what does this mean? ). If published, this will include your full peer review and any attached files.

**Do you want your identity to be public for this peer review?** For information about this choice, including consent withdrawal, please see our Privacy Policy .

Reviewer #2: No

---

## [Editor Report · Decision Letter 3]

21 Sep 2025

Perceptions of COVID-19-related nudges in the Arab world: A cross-country analysis of approval rates and associated factors

PGPH-D-25-00340R3

Dear Dr. Fakhoury,

We are pleased to inform you that your manuscript 'Perceptions of COVID-19-related nudges in the Arab world: A cross-country analysis of approval rates and associated factors' has been provisionally accepted for publication in PLOS Global Public Health.

Best regards,

Miguel Reina Ortiz, M.D., M.S., M.P.H., M.P.T., Ph.D.

Academic Editor